# The miR-200 Family of microRNAs: Fine Tuners of Epithelial-Mesenchymal Transition and Circulating Cancer Biomarkers

**DOI:** 10.3390/cancers13235874

**Published:** 2021-11-23

**Authors:** Ilaria Cavallari, Francesco Ciccarese, Evgeniya Sharova, Loredana Urso, Vittoria Raimondi, Micol Silic-Benussi, Donna M. D’Agostino, Vincenzo Ciminale

**Affiliations:** 1Veneto Institute of Oncology IOV–IRCCS, 35128 Padova, Italy; ilaria.cavallari@iov.veneto.it (I.C.); francesco.ciccarese@iov.veneto.it (F.C.); evgeniya.sharova@iov.veneto.it (E.S.); loredana.urso@unipd.it (L.U.); vittoria.raimondi@iov.veneto.it (V.R.); micol.silicbenussi@iov.veneto.it (M.S.-B.); 2Department of Surgery, Oncology and Gastroenterology, University of Padua, 35128 Padova, Italy; 3Department of Biomedical Sciences, University of Padua, 35131 Padova, Italy

**Keywords:** microRNAs, epithelial-mesenchymal transition, epithelial cancers, liquid biopsy

## Abstract

**Simple Summary:**

MicroRNAs (miRNAs) are small RNA molecules that regulate gene expression by blocking translation or inducing degradation of specific gene transcripts. The miR-200 family controls the expression of many genes that play important roles in cancer cells. One of the main pathways controlled by these miRNAs, termed epithelial-mesenchymal transition (EMT), is an essential component of the invasive growth program of solid tumors. The miR-200 family has thus been the focus of many studies aimed at discovering strategies to block cancer cell growth and disease progression. In addition, the miR-200 family miRNAs have been investigated as possible circulating cancer biomarkers. Here we provide an overview of factors that influence miR-200 family expression and target genes relevant to tumor development, followed by a summary of their potential utility as noninvasive biomarkers for selected cancers.

**Abstract:**

The miR-200 family of microRNAs (miRNAs) includes miR-200a, miR-200b, miR-200c, miR-141 and miR-429, five evolutionarily conserved miRNAs that are encoded in two clusters of hairpin precursors located on human chromosome 1 (miR-200b, miR-200a and miR-429) and chromosome 12 (miR-200c and miR-141). The mature -3p products of the precursors are abundantly expressed in epithelial cells, where they contribute to maintaining the epithelial phenotype by repressing expression of factors that favor the process of epithelial-to-mesenchymal transition (EMT), a key hallmark of oncogenic transformation. Extensive studies of the expression and interactions of these miRNAs with cell signaling pathways indicate that they can exert both tumor suppressor- and pro-metastatic functions, and may serve as biomarkers of epithelial cancers. This review provides a summary of the role of miR-200 family members in EMT, factors that regulate their expression, and important targets for miR-200-mediated repression that are involved in EMT. The second part of the review discusses the potential utility of circulating miR-200 family members as diagnostic/prognostic biomarkers for breast, colorectal, lung, ovarian, prostate and bladder cancers.

## 1. Genomic Organization of the miR-200 Family

microRNAs (miRNAs) are single-stranded noncoding RNAs (ncRNA) of about 22 nt that are essential regulators of gene expression in metazoan organisms (reviewed by [1]). The principal role of miRNAs is to post-transcriptionally regulate gene expression through binding to partially complementary sequences in specific mRNAs, most frequently in the 3′ untranslated region (3′ UTR) of the transcript. In most cases, the miRNA-mRNA interaction results in repression of the target mRNA’s translation or in its degradation [1]. However, some interactions lead to an increase in translation of the target transcript, as exemplified by the ability of miR-122 to stabilize and enhance translation of the hepatitis C virus RNA genome through binding to a 5′ UTR element (reviewed in [2]). miRNAs can also perform other functions by binding to DNA and directly regulating gene transcription, by acting as ‘sponges’ in miRNA-ncRNA-mRNA regulatory circuits, and by binding to and influencing the activity of proteins distinct from those involved in the canonical post-transcriptional silencing pathway (reviewed in [3]). Disruption of the complex mechanisms that govern miRNA expression/activity contributes to many diseases, including cancer [3].

The miR-200 family includes five evolutionarily conserved miRNAs named miR-200a, miR-200b, miR-200c, miR-141 and miR-429 that are encoded in two clusters of hairpin precursors, each of which can yield a mature -5p and -3p miRNA. In humans, the miR-200b/200a/429 cluster is located on chromosome 1p36.33, and the miR-200c/141 cluster is positioned on chromosome 12p13.31. Clustal Omega [4] alignments of the human sequences indicate 68.2–90.9% identity among the mature -3p miRNAs, while the -5p mature miRNAs show lower percent identity (63.6–81.8%). Expression data collated in the FANTOM5 database of transcriptomes from human primary cells (https://fantom.gsc.riken.jp/5/, accessed on 28 July 2021) indicate a very strong bias for production of the -3p mature miRNA compared to the -5p mature miRNA, with highest expression levels detected in epithelial cells.

The comparatively limited sequence identity and low expression levels of the miR-200 -5p miRNAs suggest that they may not be principal actors in normal epithelial cells. However, some of the -5p miRNAs, e.g., miR-200b-5p, may be functionally relevant in cancerous epithelial cells [5], and serve as cancer biomarkers [6].

The present review centers on the human -3p miR-200 family miRNAs. The miR-200 family is referred to collectively as miR-200, and the -3p suffix is omitted when citing the individual miRNAs.

The seed sequences (nt 2–8) of the human miR-200 miRNAs are identical except for an alternative U or C at the third position. This sequence variation divides the family members into 2 groups: miR-200b, miR-200c and miR-429, with U in the seed sequence, and miR-200a and miR-141, with C in the seed sequence (Figure 1A). This single nucleotide difference introduces some exclusivity for mRNA targets, an example being phospholipase C gamma 1, which is a target of miR-200b-200c-429 but not miR-200a-141 in breast cancer cells [7].

Results of expressed sequence tag (EST) mapping indicated that the miR-200b/200a/429 cluster is produced from a primary precursor (pri-miRNA) of about 6.5 or 7.5 kb [8,9], and the miR-200c/141 pri-miRNA was predicted to be about 1.1 kb in length [9]. Efforts to identify the 5′ end of the miR-200c/141 pri-miRNA identified two transcription start sites (TSS) located upstream the miR-200c hairpin, the more distal of which appears to produce a transcript that does not include the miR-141 hairpin [10]. The presence of a splice donor located between the two hairpin regions suggests that alternative splicing may also affect expression of the two miRNAs [10].

A study performed in ovarian cancer cells indicated that miR-200c and miR-141 are co-expressed with the PTPN6 gene (coding for SHP1), which is located 5′ to the miR-200c/141 locus [11]. This co-expression can occur through two mechanisms: (i) transcriptional read-through at PTPN6’s polyadenylation signal/site, resulting in production of a long transcript coding for both PTPN6 and the miR-200c/141 hairpins; (ii) formation of a DNA loop that juxtaposes the PTPN6 and miR-200c/141 promoters, resulting in similar epigenetic control [11].

## 2. The miR-200 Family and Epithelial-Mesenchymal Transition

Numerous studies have provided evidence for a critical role for miR-200 in regulating the epithelial-mesenchymal transition (EMT), an essential component of the invasive growth program of solid tumors [13,14] that is a prerequisite for the formation of metastases, the primary cause of cancer mortality. This hallmark of the cancer phenotype requires multiple steps, in which tumor cells break loose from the primary mass, invade the extracellular matrix, enter into blood or lymphatic vessels (intravasation), disseminate, leave the vascular system extravasion and establish a secondary growth at a distant site. This complex cascade of events requires the coordinated activation of an invasive growth program, which, in the context of epithelial cells, is epitomized by EMT [15]. EMT orchestrates changes in cell adhesion, motility and crosstalk of tumor cells with the stromal components resulting in invasion across physical tissue barriers. The high plasticity of these changes requires cancer cells to switch between gene expression programs driven by multiple ligand-receptor interactions (such as TGF-β-TGFR-SMAD [16], Wnt-Frizzled-β-catenin [17] and Jag-Notch [18]) that transmit intracellular signals leading to stimulation of EMT transcription factors’ (EMT-TFs) of the SNAI1 (Snail), TWIST (Twist) and ZEB families and other TFs such as NF-κB and SMAD [15,19].

Normal epithelial cells express a roster of proteins that maintain cell polarity and tight homotypic adhesion among cells and with the basement membrane. Induction of EMT-TFs leads to the repression of the expression of epithelial genes, which leads to loss of epithelial cell–cell junctions and of apical–basal cell polarity and to the activation of the mesenchymal phenotype, which includes front-to-back polarity, cytoskeletal remodeling and acquisition of cell motility [20]. In most cases, activation of the invasive program in cancer cells involves a ‘partial EMT’ with the acquisition of a limited set of mesenchymal features and retention of certain epithelial features. EMT is a highly plastic program and can be reversed to mesenchymal–epithelial transition (MET), which permits tumor cells that have arrived in a new location to establish new colonies—the final step in the invasion-metastasis process.

The miR-200 family forms part of a small army of miRNAs and other noncoding RNAs that regulate the EMT [19,21]. Key studies performed in various cell contexts established that miR-200 counteracts EMT by targeting the mRNAs coding for ZEB1 and ZEB2, favoring the expression of the epithelial marker E-cadherin, a target for transcriptional repression by ZEB [22,23,24,25,26]. miR-200 members exert direct effects on the cell phenotype by targeting mRNAs coding for cell adhesion and signaling molecules, proteases, angiogenic factors, cytoskeletal proteins and ECM components [27].

The miR-200 family also targets other drivers of the EMT transcriptional program such as TGFBR1 and SMAD2 [16], ETS1 (a transcription factor that activates ZEB1 expression [28]), the Notch ligand Jag1 and Notch transcriptional complex proteins Maml2 and Maml3 [18], PBX3 (a transcription factor upregulated by WNT signaling [29]), c-Myb (a transcription factor with oncogenic properties that promotes EMT [30]) and several co-factors in the ZEB2 and SNAI1 repressive complexes [31]. Additional miR-200 targets that can have profound effects on gene expression include QKI-5 (Quaking-1 isoform 5), an RNA-binding protein that influences mRNA splicing, trafficking, stability, and translation, SIRT1, a NAD+-dependent deacetylase with a variety of substrates that influence cell turnover, the transcription factor MYC and the long non-coding RNA (lncRNA) HOTAIR. Through its targeting of QKI-5, miR-200 members indirectly regulate alternative splicing of hundreds of transcripts during EMT [32]. The miR-200-QKI-5 functional interaction is complex, as anti-EMT and anti-tumor properties have also been described for QKI-5, suggesting that a balance between miR-200 and QKI-5 may favor the EMT vs. MET program [33]. SIRT1 deacetylates and thus modifies the activity of many proteins, including histones, with consequent repressive effects on promoters that are marked by the deacetylated histones. Studies in mammary epithelial cells showed that the SIRT1 mRNA is targeted by miR-200a [34]. In this cell system, induction of EMT resulted in an increase in SIRT1 levels paralleled by a decline in miR-200a expression, the latter effect caused in part by SIRT1-mediated repression of the miR-200b/200a/429 promoter; expression of miR-200 and SIRT1 is thus under reciprocal negative control [34]. MYC either drives or represses the transcription of a multitude of genes that regulate pathways key to cell proliferation, differentiation, senescence and death (reviewed in [35]). The MYC mRNA contains one binding site for miR-429 in its 3′ UTR and was confirmed to be a target for repression by this miRNA [36]. lncRNA HOTAIR is highly expressed in multiple cancer types, where it promotes tumor cell proliferation, invasiveness, metastasis and EMT through effects on gene transcription, epigenetic silencing, and miRNA function (reviewed in [37]). In renal carcinoma cells, miR-141 binds to HOTAIR and targets it for degradation [38].

miR-200b targets the 3′ UTR of IKBKB (inhibitor of nuclear factor kappa B kinase, subunit beta, also named IKK-beta), a protein kinase that activates NF-κB through phosphorylation of IKB. miR-200b-mediated suppression of IKBKB expression was shown to interfere with NF-κB activation in response to exposure to TNF-α [39]. Expression of JUN, a component of the AP1 transcription complex, is subjected to dual regulation by miR-200a and miR-200b [40]. These miRNAs bind to distinct sites in the JUN mRNA 3′ UTR with opposite consequences: miR-200b represses JUN expression through canonical miRNA-mediated inhibition, while miR-200a stabilizes the JUN mRNA by facilitating binding of the RNA binding protein HuR [40]. Studies of breast cancer cells demonstrated that miR-200a represses the expression of TFAM (mitochondrial transcription factor A), the principal transcription factor of mitochondria [41], an effect that hampers the proliferative capacity of cells.

The EMT phenotype is intertwined with the stem cell-like properties of cancer cells. Studies of breast cancer stem cells revealed downregulation of both miR-200 clusters in cancer cells with stem-like properties as well as in normal mammary stem cells [42]. Suppression of breast cancer cell stemness by miR-200 involves their direct targeting of mRNAs coding for BMI1 [42] and SUZ12 [43], which are components of polycomb group complexes 1 and 2 (PRC1 and PRC2), respectively. Forced expression of miR-200 members in mammary tumor cells with mesenchymal properties induces a less invasive, more slowly proliferating, partially epithelial phenotype and modifies the profile of expressed miRNAs and protein-coding genes, including many genes regulated by SUZ12 [44].

The loss of miR-200 expression in the EMT context is associated with development of resistance to wide range of chemotherapeutic agents such as microtubule-targeting taxanes [45,46], DNA-damaging agents (e.g., doxorubicin [47] and oxaliplatin [48]), the estrogen blocker tamoxifen [49] and EGFR-blocking antibodies [50]. In some experimental systems, resistance has been linked to overexpression of specific genes normally targeted by the miRNAs: for example, TUBB3, which codes for a tubulin protein known to promote resistance to taxanes, is a target of miR-200c [45]. Reintroduction of miR-200 can reverse drug resistance associated with EMT, suggesting the potential utility of miR-200 as a component of chemotherapy regimens [45,46,47,50]. Investigations of mouse models of breast cancer with inducible miR-200 expression highlighted the miRNAs’ ability to block disease initiation [51] and metastasis [52], findings that support their possible application as drugs for breast cancer prevention and treatment. Figure 1B illustrates important miR-200 family-target gene interactions and their biological consequences.

The potential for miRNAs to circulate in biological fluids after release from cells in association with extracellular vesicles, Argonaute proteins or high-density lipoproteins (HDL) expands their range of effects to neighboring cells or even to distant tissues (reviewed in [53]). This is exemplified by a study of colon tumor cell spheroids [54] which showed that miR-200-containing exosomes released by the spheroids inhibited an EMT like-program in an underlying layer of lymph endothelial cells (LEC), which would otherwise facilitate the process of tumor intravasation in lymphatic vessels. Interestingly, the extent of LEC, EMT was directly related to the degree of resistance to 5-fluorouracil of the tumor spheroids; highly chemoresistant tumor subclones lost expression of miR-200 and no longer provided anti-EMT signals to the LEC [54].

The connection between miR-200 and EMT suppression is consistent with the definition of these miRNAs as tumor suppressors. The biological effects of individual miR-200 members can differ widely—for example, investigations of pancreatic cancer cells ectopically expressing miR-200 family members identified miR-429 as most effective in limiting in vivo tumor cell growth and metastatic dissemination [55]. It is also noteworthy that miR-200 members can target mRNAs coding for tumor suppressors, a property that would define them as ‘oncomiRNAs’: an important example is found in endometrial carcinoma cells, in which miR-200a, miR-200b and miR-429 were shown to directly target the mRNA coding for PTEN, a well-known tumor suppressor [56,57].

## 3. Factors That Control miR-200 Family Expression and Function

Given their importance in controlling EMT and cancer progression, a thorough understanding of the upstream cues that govern miR-200 expression is of critical importance for designing anticancer strategies based on rewiring these regulatory circuits. An investigation of the factors that suppress miR-200 expression during EMT revealed the presence of ZEB-type E-box elements upstream of the TSS of the miR-200b/200a/429 primary precursor RNA (pri-miRNA), and demonstrated that the expression of the cluster is repressed by ZEB1 and ZEB2 [8]. The miR-200c/141 locus was likewise found to contain ZEB-family responsive elements that mediate repression of miR-200c-141 production by ZEB1 [58]. Thus, ZEB1/2 and miR-200, which exert opposite functions on the EMT, reciprocally regulate each other in a double negative feedback loop [8,13,58]. FOXM1B, a member of the Forkhead Box family of transcription factors, represses transcription from the promoters of both miR-200 clusters [59]. Both promoters are also repressed by the transcription factor NANOG, an important driver of self-renewal in pluripotent cells and of EMT in colorectal cancer cells [60]. The stemness and growth properties of colorectal cancer cells are influenced by a reciprocal feedback loop involving miR-200c and the homeodomain transcription factor SOX2, with miR-200c targeting the SOX2 mRNA, and SOX2 repressing the miR-200c promoter [61].

Activation of the HIF-1 transcription complex e.g., by hypoxia, in colorectal cancer (CRC) cells triggers upregulation of the transcription factor ASCL2 [62]. Binding of ASCL2 to the promoter region of the miR-200b/200a/429 locus represses its expression and thus supports EMT [62,63]. In turn, miR-200b directly represses expression of the HIF1A mRNA to form a HIF1A-ASCL2-miR-200 feedback loop that contributes to the ability of CRC cells to switch between EMT and MET [62]. Recent experiments that employed a miR-200b/c-sensitive fluorescent protein sensor reinforced the role for miR-200 in impeding EMT while promoting cell proliferation and implicated metabolic signaling in the control of miR-200b/c expression [64].

A major contribution to the plasticity of the EMT/MET phenotype is conferred by reversible DNA methylation of CpG islands in the miR-200 loci that results in their transcriptional repression [65]. The promoter regions of both miR-200 clusters contain CpG islands that undergo DNA methylation during EMT in diverse cell types [10,34,66,67,68,69]. Experiments performed in gastric cancer cell lines and a glioblastoma cell line indicated that epigenetic silencing of the miR-200b/200a/429 promoter is mediated by CpG methylation catalyzed by DNMT1 (DNA methyltransferase 1) and EZH2-directed histone 3 trimethylation (H3K27me3), with recruitment of DNMT1 depending on EZH2 [70]. A study of triple-negative breast cancer cells revealed a positive correlation between levels of ZEB1 and silencing of the miR-200c/141 promoter by CpG methylation and histone H3K9 trimethylation (H3K9me3, another mark of epigenetic silencing) [71].

Repression of the miR-200b/200a/429 promoter by MYC was documented in endometrial cancer cells [72] and in triple-negative breast cancer cells [73]. In triple-negative breast cancer cells, binding of MYC to the miR-200b/200a/429 promoter leads to recruitment of DNMT3A to promoter CpGs, resulting in promoter methylation [73]; negative feedback control is provided by direct targeting of the DNMT3A mRNA by miR-200b [73]. A similar feedback relationship connects miR-200 to FERMT2 (also named Kindlin-2), an adaptor protein of the 4.1-ezrin-radixin-moesin (FERM) domain-containing protein family that interacts with integrins and promotes formation of invadopodia and remodeling of the extracellular matrix, which are characteristic events of EMT. Studies of breast cancer cells revealed a further role for FERMT2 in controlling EMT by interacting with DNMT3A and promoting CpG methylation of both clusters’ promoters [74]. An investigation of the role of FERMT2 breast cancer metastasis showed that it is a direct target of miR-200b; forced expression of miR-200b in a breast cancer cell line expressing high levels of FERMT2 reduced the tumor-forming ability and metastatic properties of the cells when injected into mice, and led to downregulation of several EMT markers [75]. The histone demethylase KDM5B can repress expression of both miR-200 clusters through demethylation of histone H3K4me3 (a marker of transcriptionally active genes) in the clusters’ promoter regions [76]. In breast cancer cells, PELP1, a coregulator of nuclear receptors that is overexpressed in several hormone-driven cancer types, represses expression of both miR-200 clusters by recruiting histone deacetylase 2 to the loci’s promoter regions [77]. In prostate cancer cells, the transcriptional repressor ZBTB33 (also named Kaiso) participates in downregulation of miR-200 family expression induced by EGF signaling through direct interaction with methylated regions of the miR-200 promoters [78].

An investigation of miRNAs affected by RAS signaling showed that both miR-200 clusters are downregulated in the presence of oncogenic KRAS (KRASG12D), with consequent promotion of cell survival and EMT [79]. This repressive effect was attributed to the combined action of ZEB1 and transcription factors JUN and SP1 (downstream components of RAS signaling), which were shown to interact with specific sites in the miR-200 promoters [79].

Other studies indicated that SP1 drives expression of the miR-200b/200a/429 cluster [80]. The nearly ubiquitous expression of SP1 led to the proposal that SP1 maintains miR-200b/200a/429 expression and the epithelial phenotype in the absence of ZEB [80]. Other transcription factors that contribute to activate expression of the miR-200b/200a/429 cluster include Smad3, in gastric cancer cells [81] and ERG, in prostate cancer cells [82]. ERG is an ETS-family transcription factor that is frequently upregulated in prostate cancer as a result of a translocation involving the TMPRSS2 gene [82]. ERG-mediated activation of miR-200b/200a/429 expression interferes with prostate cancer proliferation and invasiveness, an effect that implicates the ERG-miR-200 interaction in the indolent behavior of prostate cancer in the majority of patients [82].

The promoters of both miR-200 clusters are activated by another ubiquitously expressed basic transcription factor named KLF5 [83], and by c-Myb [84]. c-Myb expression was found to correlate with levels of miR-200a, miR-200b, miR-200c and miR-141 in breast cancer samples, and transfection experiments showed that the repressive effects of ZEB1 on miR-200 expression dominate over the activating effects of c-Myb [84]. As mentioned above, c-Myb is a target for repression by miR-200b, miR-429 and miR-200c, thus adding c-Myb to the miR-200-ZEB feedback pathway that governs EMT.

The promoter regions of both miR-200 loci also contain binding sites for p53. Accordingly, p53 can upregulate the expression of both miR-200 clusters [85,86]. The p53 family members p63 and p73 were also shown to activate the miR-200b/200a/429 locus [87]. In ovarian cancer cells, the transcription factor GRHL2 (grainyhead-like 2) was shown to favor the epithelial phenotype by increasing miR-200b/200a/429 expression through direct transcriptional activation and by reducing the levels of the repressive histone H3K27me3 mark in the cluster’s promoter and CpG island [88].

Studies of breast cancer cells [89] indicated a connection between EMT, miR-200 and TP53BP1 (p53 binding protein 1), a tumor suppressor protein involved in repair of double-strand DNA breaks (reviewed in [90]). Forced expression of TP53BP1 in breast cancer cell lines led to upregulation of miR-200b and miR-429, downregulation of ZEB1, and an epithelial-like phenotype, while silencing of TP53BP1 had the opposite effects [89]. An analysis of a small panel of breast cancer samples confirmed the positive correlation between the expression levels of TP53BP1 and the two miRNAs and the negative correlation between levels of TP53BP1 and ZEB1. The levels of TP53BP1 were significantly higher in tumor samples from patients without lymph node metastasis compared to those with lymph node metastasis [89]. The mechanism through which TP53BP1 influences miR-200b/miR-429 expression, and how this protein fits into the miR-200b/ZEB feedback loop, remain open questions.

miR-200 expression is activated by several nuclear receptor pathways in different cell contexts. Analyses of prostate cancer cell lines identified miR-141 [91,92], miR-200a [92,93], miR-200b [93] and miR-200c [93] among many miRNAs that are upregulated upon stimulation of androgen receptor (AR) signaling by dihydrotestosterone and the synthetic androgen R1881. In liver cancer cells, miR-200c is activated by nuclear receptors PPARα (peroxisome proliferator activated receptor alpha) and LRH-1 (liver receptor homolog-1), and is inhibited by SHP (small heterodimer partner), a transcriptional repressor that interacts with PPARα and LRH-1 [94].

Many lncRNAs have been identified as post-transcriptional regulators of miR-200 abundance/function, primarily by acting as competitive endogenous RNAs (ceRNAs). Examples are lncRNA-ATB, MALAT1 and H19, each of which has been described to regulate miR-200 in multiple cancer types. Studies of hepatocellular carcinoma (HCC cells) [95] revealed the ability of lncRNA-ATB (activated by TGF-β) to promote EMT, invasion and metastasis through its ceRNA activity against miR-200 members, which subtracts (sponges) the miRNAs from mRNAs coding for ZEBs and other EMT-promoting factors. Analogous ceRNA activity was described for MALAT1 [96]. The relationship between H19 and miR-200 is more complex: in gastric cancer cells, miR-141 and H19 were shown to compete with each other for mRNA targets, whereas in HCC, H19 upregulates miR-200 family expression through a mechanism involving increased histone acetylation [97].

miR-200 abundance and activity are also regulated post-transcriptionally at the pri-miRNA processing stage and through editing of the miRNA sequence by adenosine deaminases acting on RNA (ADARs). Studies of ovarian cancer cells demonstrated that expression of both miR-200 clusters is influenced by the RNA binding protein DDX1 through its recruitment of the pri-miRNAs to the Drosha microprocessor (the complex that releases the pre-miRNA hairpin) [98]. ADARs are a family of enzymes that convert adenosines to inosines, which are recognized as guanosines during base-pairing. In thyroid cancer cells, miR-200 members undergo substantial A-to-I editing by ADAR1, a modification that impairs the ability of the miRNAs to target the ZEB1 mRNA [99]. A recent investigation of the impact of the protein kinase PKCζ in CRC cells revealed a role for PKCζ and ADAR2 in regulating the intracellular levels of miR-200 [100]. Experiments carried out in a CRC cell line demonstrated that loss of PKCζ (a tumor suppressor that is downregulated in CRC metastases) is associated with downregulation of miR-200 members caused by an increase in their secretion in extracellular vesicles, a situation that favors EMT. ADAR2, whose RNA editing activity depends on phosphorylation by PKCζ, promotes retention of miR-200 inside the cell, thus favoring the epithelial phenotype [100]. Consistent with the role of miR-200 in EMT, loss-of-function of the PKCζ/ADAR2 axis activated the EMT program and increased liver metastases in a mouse xenograft model [100]. In accordance with these findings, analyses of CRC patients’ data in TCGA showed that low expression of miR-200b and miR-200a correlated with substantially shorter overall survival; furthermore, the levels of the mRNA coding for PKCζ correlated with levels of miR-200b/200a/429 [100]. The discovery of this mechanism controlling intracellular miR-200 levels provides an explanation for the highly metastatic properties of PKCζ-deficient CRC cells. The exact role of ADAR2 in maintaining miR-200 inside cells remains to be identified, and the possibility that PKCζ/ADAR2-mediated control of miR-200 retention/elimination operates in other cancer types remains to be determined. Figure 2 depicts the main regulatory networks involving miR-200.

## 4. Circulating miR-200 as Biomarkers

The evaluation of miRNAs in biological fluids (liquid biopsy) to detect cancer was initiated with studies that identified serum miR-21 as a potential diagnostic/prognostic marker for diffuse large B-cell lymphoma [101] and serum miR-141 as a potential diagnostic marker for prostate cancer [102]. These observations, together with the property of extracellular miRNAs to resist degradation by RNase A [103], opened the door to a rich body of research on circulating miRNAs and other noncoding RNAs as diagnostic and prognostic markers for cancer patients detectable with a minimally invasive sampling procedure (reviewed in [104,105]). The following sections provide an update on the status of extracellular miR-200 as biomarkers in a selection of clinically important epithelial-derived neoplasms.

### 4.1. Breast Cancer

Female breast cancer (BC) is the most frequently diagnosed cancer worldwide [106]. The molecular subtypes of breast cancer are defined by their expression of estrogen and progesterone receptors (HR), and epidermal growth factor receptor 2 (HER2) amplification. HR+/HER2− (Luminal A) breast cancers are the most common subtype, followed by the triple-negative (TNBC), HR+/HER2+ (Luminal B) and HR−/HER2+ (HER2+) subtypes, in that order [107]. TNBC is the most aggressive subtype, with an enrichment of stem-like tumor cells [108].

The miR-200 family forms part of a complex network of miRNAs that determine the invasive properties of breast cancer cells (reviewed in [109]), with low miR-200 levels favoring EMT [26], and higher levels promoting the MET phenotype, which permits tumor cells that have escaped the primary tumor to establish colonies in a new location [110,111]. Experiments performed on panels of murine and human breast cancer cell lines with differing metastatic properties showed that highly metastatic cells with high levels of miR-200 produce miR-200-containing extracellular vesicles (EVs) [112]. These miR-200-containing EVs are taken up by poorly metastatic cells, with resulting alterations in gene expression supportive of MET and an increase in their metastatic growth [112]. Exosome-packaged miR-200 might therefore act as a hormone-like modulator of MET. The contribution of miR-200 to metastatic progression is clinically relevant, given the fact that many breast cancer patients eventually die of metastatic disease despite early diagnosis and good initial response to therapy.

On the whole, investigations of circulating miRNAs as biomarkers in blood, serum or plasma to distinguish breast cancer patients from healthy controls have given mixed results, with limited overlap in panels of identified miRNAs [reviewd in [113,114,115]. Initial investigations that employed single-target RT-PCR or array profiling either did not examine miR-200 [116,117] or did not identify miR-200 among miRNAs with different abundance in patients vs. controls [118,119,120,121,122]. A next generation sequencing (NGS)-based analysis of pooled serum samples from breast cancer patients and controls also did not indicate differential expression of miR-200 [123]. However, another NGS analysis of serum samples yielded a list of 26 upregulated miRNAs and 17 downregulated miRNAs in patients with stage I or II disease compared to healthy controls; upregulated miRNAs included miR-200b, miR-200c and miR-429, and downregulated miRNAs included miR-200a [124]. In contrast, an analysis of miR-200c and miR-141 in unfractionated blood samples detected reduced levels of miR-200c in breast cancer patients (early-stage grouped together with advanced-stage) compared to controls (levels of miR-141 did not differ significantly in the 2 groups) [125]. An exploratory analysis of a panel of 9 miRNAs that included miR-200b and miR-200c in urine samples from breast cancer patients and controls did not reveal significant differences in levels of these miRNAs in patients vs. controls [126].

In contrast to the lack of a consensus supporting circulating miR-200 for early diagnosis of breast cancer, several analyses have implicated circulating miR-200 as marker of metastatic dissemination, a role that first emerged from studies of circulating tumor cells (CTC) derived from the primary tumor mass that are detected in the circulation. A subpopulation of CTC is considered to produce metastatic disease, and CTC analysis has become an important tool for predicting prognosis of metastatic disease in several solid cancer types, including breast cancer (reviewed in [127]). A search for plasma miRNAs that could serve as surrogate markers for CTC in patients with metastatic breast cancer identified a panel of 8 miRNAs that included miR-200a, miR-200b, miR-200c and miR-141 whose plasma levels were higher in CTC (+) patients compared to CTC (−) patients or healthy controls [128]. Further statistical analyses indicated that plasma levels of miR-200b distinguished CTC (+) from CTC (−) patients and yielded predictions of PFS (progression free survival) and OS (overall survival) that matched or surpassed the performance of CTC enumeration [128]. In accordance with these findings, higher levels of miR-200b (and miR-7) were detected in blood samples from patients with lymph node-positive breast cancer compared to patients without lymph node involvement, and in blood of patients with distant metastases compared to patients without metastases [111]. Another analysis of circulating miR-200c and 3 other miRNAs (miR-21, miR-146a and miR-210) confirmed increased levels of miR-200c in CTC and plasma from metastatic breast cancer patients compared to healthy controls, but did not link circulating miR-200c levels to diagnosis or prognosis [129].

Xenograft experiments performed with a highly metastatic human triple-negative inflammatory breast cancer (IBC) cell line revealed a role for miR-141 in metastatic colonization in the brain [130]. An analysis of miR-141 in sera from a cohort of 105 breast cancer patients showed that patients with metastatic inflammatory breast cancer (IBC) had elevated serum levels of miR-141 compared to those with locally advanced disease or metastatic non-IBC. In metastatic patients, detectable serum miR-141 levels were associated with a shorter time interval before development of brain metastasis and with considerably shorter PFS and OS [130].

Studies of mice with reduced expression of the transcription factor FOXP3, which leads to breast cancer with lung metastases in females, provided further evidence for exosome-mediated release of miR-200 members from tumor cells and their utility as markers of metastasis [131]. The miR-200c/141 locus was identified as an indirect target of FOXP3-mediated activation [131]. While primary breast tumor cells and lung metastases showed reduced expression of miR-200c/miR-141 compared to normal breast epithelial cells, the plasma levels of miR-200c and miR-141 increased during disease progression, with increasing levels of exosome-associated miR-200c and miR-141 most evident in mice with metastatic disease [131]. Interrogation of a TCGA cohort of breast cancer patients identified a positive correlation between levels of FOXP3 and miR-200 members in breast cancer samples, and reduced expression of miR-200 in tumors from patients with metastatic disease. On the other hand, the levels of miR-200c and miR-141 were higher in plasma from patients with metastatic breast cancer tissue compared to patients with localized disease, benign breast tumors or disease-free controls [131].

Kaplan–Meier analyses of miR-200c and miR-141 expression levels, clinical data and tumor characteristics indicated a significant association between high blood levels of miR-200c and/or low levels of miR-141 and poor PFS and OS independent of disease stage or hormone receptor status, suggesting the utility of these miRNAs as prognostic indicators [125]. An exploratory RT-PCR-based profiling assay identified sets of miRNAs whose high plasma levels were associated with poor OS and PFS of metastatic breast cancer patients; both sets included all of the miR-200 family members. Further evaluations showed that a miRNA panel consisting of miR-141, miR-200a, miR-200b, miR-429 and miR-1274a (all high) performed better than CTC enumeration for predicting overall survival [128,132].

Metastatic disease frequently arises in breast cancer patients many years after surgical intervention and adjuvant chemotherapy, a feature that contributes to patient anxiety and complicates follow up. A study by Madhavan et al. [132] revealed an association between plasma levels of a 6-miRNA panel that included miR-200a, miR-200b and miR-200c, and development of metastases as early as 2 years prior to clinical diagnosis of metastatic spread.

Papadaki et al. [133] investigated whether levels of a panel of miRNAs (miR-21, miR-23b, miR-190, miR-200b and miR-200c) measured in plasma samples after surgical intervention and before adjuvant chemotherapy, correlated with relapse and survival of early breast cancer patients. Results showed that patients who relapsed had higher plasma levels of miR-200c, miR-21 and miR-23b, and lower plasma levels of miR-190, compared to those who did not relapse. miR-200c and miR-21 were more abundant in plasma of patients who relapsed 5 or more years after diagnosis (late relapse) compared to patients who remained disease-free during the follow up period. Elevated plasma levels of miR-200c and miR-21 were each associated with shorter disease-free survival (DFS), and patients with elevated plasma levels of both miRNAs had shorter DFS compared to those with one elevated miRNA. Elevated miR-21 (but not miR-200c) was also associated with shorter OS. The combined evaluation of plasma miR-200c levels, axillary lymph node infiltration, tumor grade and ER status predicted late relapse with an AUC of 0.890 (sensitivity 75%, specificity 89%) [133]. A subsequent study by Papadaki et al. [134] evaluated miR-21, miR-23b, miR-190, miR-200b and miR-200c in plasma from patients with early breast cancer or metastatic disease [134]. Higher levels of miR-21, miR-23b, miR-200b and miR-200c were detected in patients with metastatic disease. Levels of miR-21 and miR-200b were also higher in premenopausal metastatic patients compared to postmenopausal metastatic patients. Levels of miR-21 and miR-200b each distinguished patients with metastatic vs. early disease, but higher accuracy was obtained by combining miR-21, miR-200b, miR-190 and miR-200c. High levels of miR-200b predicted shorter OS in patients with metastatic disease and in HER2-negative patients [134]. These findings underscore the link between miR-200 and metastatic spread and indicate them as markers for early detection of metastatic disease and prognosis. However, this assertion is tempered by results of recent investigation of serum markers of breast cancer that included miR-200 members in its list of interrogated miRNAs, which indicated an association between high serum levels of miR-141 and better survival [135]. This study also proposed a 5-miRNA panel for serum-based detection of breast cancer which however did not include miR-141 or other miR-200 members [135].

### 4.2. Lung Cancer

Lung cancer is the second most frequently diagnosed cancer and the most frequent cause of cancer deaths worldwide [106]. Lung cancer comprises a group of aggressive malignancies of epithelial derivation (non-small cell lung carcinoma, NSCLC, 85% of the cases) or neuroendocrine derivation (small cell lung carcinoma, SCLC, 15% of the cases). Based on its histological features, NSCLC is further divided into three major subtypes: lung adenocarcinoma (LAD) (40%), squamous cell carcinoma (SCC) (25–30%) and large cell carcinoma (5–10%) [136].

Distant metastases represent the first cause of NSCLC-related death, and 30–40% of NSCLC patients have metastatic disease at the time of diagnosis [137]. Targeted therapies, consisting of tyrosine kinase inhibitors (TKI) for oncogene-addicted NSCLC or immuno-checkpoint inhibitors (ICIs) for non-oncogene-addicted tumors with elevated expression of programmed cell death 1 (PD-L1), have significantly improved the outcome of NSCLC patients, although the prognosis of stage 4 patients remains dismal. Reliable biomarkers are thus needed to facilitate early diagnosis and to optimize the different therapies for individual patients [138]. To this end, several studies investigated the possible role of miR-200 as novel diagnostic tools in NSCLC (reviewed in [139]).

Studies of miR-200 in tumor tissues have yielded conflicting results. Consistent with the role of miR-200s in EMT, an analysis by Ceppi et al. of 69 NSCLC tissues (73% LAD and 27% SCC), showed that low miR-200c expression correlated with tumor spread to distant lymph nodes [67]. These observations were supported by analyses of LAD samples which indicated a negative correlation between miR-200c and clinical stage [140], and positive correlation between miR-200b and OS [141]. In contrast with these findings, Tejero et al. showed that increased expression of miR-200c and miR-141 in LAD tissues (*N* = 73) correlated with shorter OS [142], and Si and colleagues showed a positive correlation between high miR-200c expression and lymph node metastases/poor OS [143].

These discrepancies might reflect changes in expression of this miRNA family according to the EMT status. As reported above, EMT is a highly plastic process in which cells from the invasive front of primary tumors acquire mesenchymal features needed for distant spread. At the metastatic foci, seeded tumor cells need to switch from EMT to the MET program and re-acquire epithelial features to complete the metastatic process [144]. Thus, the plasticity of the EMT program may result from differential expression of the miR-200 family members, closely related to the specific tumor site observed (invasive front vs. core of primary tumors, circulating tumor cells and metastatic colonies). These considerations suggest that determination of miR-200 expression in tumor biopsies may provide information that is of excessive complexity for diagnostic/prognostic purposes.

Liquid biopsy might better represent the intratumor heterogeneity and temporal evolution of the invasive program of lung cancer. This is supported by the findings of increased levels of cell-free miR-200b and miR-141 in the plasma/serum and sputum of early-stage LAD patients (compared to healthy controls) [6,145,146,147] and higher levels of miR-200b, miR-200c and miR-141 in pleural effusions of LAD patients (compared to patients with lung inflammatory disease) [148].

A possible connection between miR-200/EMT and immune checkpoints was revealed by studies demonstrating that miR-200 members directly target PD-L1 and by an analysis of TGCA datasets of early-stage LAD, showing a strong positive correlation between EMT markers and PD-L1 and a strong negative correlation between PD-L1 and miR-200 [149,150]. Fan et al. showed that patients with advanced NSCLC who exhibited a good response to the immune checkpoint inhibitor nivolumab had higher serum levels of miR-200 (the authors did not indicate the specific miRNA of this family) compared to non-responder patients [151]. In addition to contributing to our understanding of the mechanism regulating PD-L1 in cancer, further investigation of the link between miR-200 and PD-L1 expression might yield new biomarkers to select patients who will respond to PD-L1 inhibitors.

### 4.3. Colorectal Cancer

Colorectal Cancer (CRC) is the third most commonly diagnosed malignancy worldwide and the second leading cause of cancer death [106]. CRC is a heterogeneous disease that is classified on the basis of histological features and driver mutations. Stage I and II CRC tumors are curable by surgery, and more than 70% of stage III patients are successfully treated by combining surgical excision and adjuvant chemotherapy. Unfortunately, despite recent improvements in chemotherapy, patients with stage IV disease remain incurable in most cases [152]. Therefore, the discovery of novel tools for the early diagnosis and prognostic stratification of CRC patients is of key importance to improve the outcome of these patients.

A recent systematic review of the role of miR-200 in EMT of CRC cells and their possible utility as biomarkers highlighted the poor agreement among studies that examined miR-200 expression in tumor samples, with levels of individual miR-200 members showing upregulation, downregulation, or no substantial change in CRC samples vs. controls [153]. These inconsistencies may in part reflect the contribution of variable amounts of stromal cells in the tumor samples and in heterogeneity of the tumor cells in different areas of the tumor mass. Indeed, several studies reported downregulation of miR-200b and/or miR-200c at the invasive front of CRC tumor masses that broached the basement membrane [154,155,156].

Considerable effort has been invested to identify CRC markers analyzable through liquid biopsy (reviewed in [157]). A survey of studies on circulating miRNAs in CRC identified miR-21 as the most frequently proposed candidate for CRC diagnosis, and miR-200a, miR-200b, miR-200c and miR-141 as most frequently connected to CRC prognosis [158]. One of the first investigations of circulating miRNAs in CRC diagnosis detected increased plasma levels of miR-200a, miR-200b and miR-200c (and many other miRNAs) in CRC patients vs. controls in the discovery phase of the study, but did not measure these miRNAs in the validation cohort, and proposed miR-92 as a marker for early CRC detection [159]. A subsequent evaluation of miR-200c and miR-18a (chosen on the basis of their upregulation in a set of CRC tumor samples) in plasma samples from CRC patients and controls showed upregulation of both miRNAs in the majority of the patients’ samples, and a reduction in their levels after tumor resection [160]. In ROC analysis the combination of both miRNAs yielded an AUC of 0.839 for patients vs. controls [160]. An evaluation of an 8-miRNA panel that included miR-200c measured lower levels of miR-200c in plasma and tumor tissue from CRC patients (stages II, III and IV) compared to control samples and did not identify miR-200c as a diagnostic biomarker [161].

An analysis of miR-141, miR-21 and miR-92 in plasma from large cohorts of CRC patients and controls detected a reduction in plasma miR-92 and no significant difference in plasma miR-21 or miR-141 levels in CRC patients vs. controls [162]. However, stratification of the patients revealed higher plasma levels of miR-141 in stage IV disease compared to lower stages and controls, and identified miR-141 as a predictor of poor prognosis [162]. Moreover, evaluation of plasma miR-141 plus carcinoembryonic antigen (CEA) improved the accuracy of detection of stage IV CRC compared to either marker alone [162].

A discovery-validation study of plasma miRNAs that included a validation cohort of 187 CRC patients and 47 controls identified a 4-miRNA panel consisting of miR-96, miR-203, miR-141 and miR-200b with potential value for CRC prognosis [163]. ROC analysis indicated that high plasma miR-141 levels accurately distinguished stage IV patients from stage I-III patients (AUC, 0.851), and univariate survival analysis indicated an association between lower plasma levels of miR-96 and miR-200b and better overall survival [163].

In another discovery-validation study, a miRNA-microarray profiling analysis identified miR-141 among many upregulated miRNAs in sera from CRC patients compared to controls, and RT-PCR assays confirmed upregulation of miR-141 and 3 other miRNAs (miR-31, miR-224-3p and miR-576-5p) and downregulation of miR-4669 in sera from patients compared to controls [164]. Levels of miR-141, miR-31, miR-224-3p and miR-576-5p were also higher in sera from stage III/ stage IV patients compared to those with stage I or II disease. ROC curve analysis supported the potential utility of the 5-miRNA panel for CRC diagnosis [164]. On the other hand, another recent investigation aimed at determining whether plasma levels of miR-141, miR-221-3p and miR-222-3p might be useful for diagnosis of localized CRC indicated that miR-141 was not significantly upregulated in patients vs. controls [165]. Interestingly, among CRC patients with localized disease, increased plasma levels of miR-141 and miR-425-3p were found to be associated with the presence of tumors harboring activating mutations in KRAS [165], a genetic alteration that is known to correlate with more aggressive disease [165,166].

The clinical importance of miR-200 as circulating biomarkers for metastatic CRC was supported by two studies of ample patient cohorts (>400 subjects) [167,168]. Results of a three-step study to evaluate the potential utility of miRNAs controlling EMT as serum biomarkers in CRC indicated that the serum levels of miR-200c were significantly higher in stage IV CRC patients compared to stage I-III patients [167]. Serum miR-200c positively correlated with lymph node metastasis, distant metastasis and prognosis, and was an independent predictor for lymph node metastasis (odds ratio: 4.81, *p* = 0.0005) and tumor recurrence (hazard ratio: 4.51, *p* = 0.005) and an independent prognostic marker (hazard ratio: 2.67, *p* = 0.01) [167]. Maierthaler et al. [168] investigated the plasma levels of circulating miRNAs in 543 stage I-IV CRC patients, with candidate miRNAs initially discovered through a profiling assay on plasma from 10 non-metastatic and 10 metastatic cases, followed by validation of a selected panel of miRNAs, including all of the miR-200 family members, in the full sample cohort. Results showed that patients with metastatic disease exhibited a modest but significant increase in the levels of circulating miR-141, miR-200a, miR-200b, miR-203a and miR-122 compared to patients without metastases, and elevated miR-122 correlated with poor prognosis [168]. An evaluation of serum levels of miR-200b, miR-200c and miR-141 from 139 CRC patients at different disease stages indicated significant elevation of the 3 miRNAs in stage-IV patients (all of whom had liver metastases) compared to earlier-stage patients [169].

Studies of exosome-associated miR-200 levels in CRC patients have produced conflicting results. A microarray-based discovery analysis of serum exosome-associated miRNAs did not detect upregulation of any of the miR-200 members in 88 CRC patients (stages I through IV) vs. 11 controls [170]. A more recent examination of 10 selected miRNAs in extracellular vesicles isolated from sera of 44 CRC patients with metastatic disease and 17 healthy controls identified 7 miRNAs, including miR-200b, that were significantly more abundant in patients compared to controls, but miR-200b did not pass statistical tests for association with diagnosis or prognosis [171]. A comparison of all 5 miR-200 members in the plasma and exosome fractions of blood drawn from a peripheral vein (PV) or from the tumor-draining mesenteric vein (MV) revealed higher levels of the miRNAs in MV samples compared to the PV samples, and an association between low MV plasma/exosome miR-141 and miR-200c and longer survival [172].

Serial measurements of Ago2-associated miR-21 and miR-200c in plasma from CRC patients with metastatic disease receiving cycles of systemic chemotherapy revealed fluctuations in the levels of the 2 miRNAs [173]. The investigators proposed that the spikes in Ago2-miRNA levels were a consequence of massive cell death in the metastatic lesions, and that such measurements might provide an indication of response to treatment [173].

### 4.4. Prostate Cancer

Prostate Cancer (PC) is a clinically variable and molecularly heterogeneous disease that represents the second most frequently diagnosed tumor in males [106]. Testosterone is necessary for the replication and survival of normal prostate cells and androgen signaling plays a key role in both induction and progression of PC. However, some PC patients already have androgen-independent cancer at diagnosis, or become androgen-independent (also defined as castration-resistant PC, CRPC) after the start of androgen deprivation therapy (ADT). Constitutive activation of androgen receptor (AR) signaling, a key alteration in CRPC, may result from expression of the AR-V7 splice variant of the AR, hypersensitivity to androgens, overexpression of AR and intratumoral steroidogenesis [174,175]. As noted in Section 3, miR-200a, miR-200b, miR-200c and miR-141 are upregulated upon activation of androgen receptor signaling in prostate cancer cells.

Various studies reported heterogeneity in expression of miR-200 members in prostate cancer cell lines, cancer tissue and biological fluid samples [176].

As mentioned at the beginning of Section 4, one of the first analyses of circulating miRNAs in cancer patients revealed significantly higher levels of miR-141 in sera from 25 patients with metastatic PC compared to 25 healthy controls [102] and several subsequent studies confirmed circulating miR-200 as an indicator of metastatic PC. A miRNA profiling analysis of sera from 14 PC patients with localized disease and 7 bone-metastatic PC patients identified miR-141, miR-200b and miR-200c among upregulated miRNAs in patients with metastatic disease [177]. Further validation assays on additional PC patients of high- and intermediate-risk groups identified elevated levels of miR-141 and miR-375 as the best markers to distinguish patients with high-risk disease (i.e., those with Gleason score ≥ 8 or lymph node involvement) from patients with intermediate-risk disease (Gleason score = 7 or N0). While both miRNAs correlated with lymph node involvement, only miR-141 was significantly upregulated and correlated with Gleason score. The investigators also found a highly significant upregulation of miR-141 and miR-375 in PC tissues compared to control prostate epithelium samples [177]. Another investigation of serum samples from patients with localized or metastatic PC confirmed upregulation of miR-141 and miR-375, along with miR-378, in the metastatic patients, as well as upregulation of miR-141 and miR-375 in primary tumor samples compared to control prostate tissue [178].

An analysis of plasma levels of miR-21, miR-141 and miR-221 in a cohort of 51 PC patients showed that miR-21 and miR-221, but not miR-141, were significantly more abundant in patients compared to controls [179]. However, stratification of the patients revealed that all 3 miRNAs were more abundant in plasma from patients with metastatic disease compared to those with localized tumors or early-advanced disease, with miR-141 showing the strongest difference [179]. A two-step discovery/validation analysis of serum pools from 25 metastatic CRPC patients and 25 healthy donors (discovery step) and 21 metastatic CRPC patients and 20 controls (validation step) identified high levels of miR-200a, miR-200c, miR-141 and 2 other miRNAs (miR-210 and miR-375) in sera from patients compared to controls; ROC analysis identified miR-141 as the most accurate miRNA for discriminating metastatic CRPC patients from controls, with an AUC of 0.899 [180]. In another cohort of patients with localized PC or mCRPC, miR-141 alone did not discriminate between the 2 patient groups, but combined measurement of miR-141 and 2 other miRNAs (miR-151-3p and miR-16) plus PSA dosage differentiated the patients with high sensitivity and specificity, yielding a ROC AUC of 0.968 [181]. A study of microvesicle-associated miRNAs in plasma identified increased levels of miR-141, miR-200b and miR-375 in patients with metastatic PC compared to patients with localized PC; analysis of microvesicles from sera of another patient cohort confirmed higher miR-141 and miR-375 levels in the patients with metastatic PC [182].

Further testing of miR-141 as a marker of bone-metastatic PC confirmed its increased serum levels in patients with metastatic bone lesions compared to patients with localized PC or benign prostatic hyperplasia (BPH) [183]. Serum miR-141 levels correlated with Gleason score, the number of metastatic lesions, and circulating levels of alkaline phosphatase (a marker of bone remodeling and bone lesions) in the metastatic patients, but did not correlate with serum PSA [183]. Analysis of an 8-miRNA panel that included miR-200b and miR-200c in plasma samples from 102 PC patients and 50 controls identified increased levels of miR-200c and reduced levels of miR-200b in the patient group, and proposed the 2 miRNAs together with 2 mRNAs (OR51E2 and SIM2) as diagnostic/prognostic markers [184].

A study aimed at identifying circulating miRNAs able to predict response of CRPC patients to docetaxel analyzed plasma or serum samples from a cohort of 97 patients before and after treatment [185]. Results indicated that non-responders (patients whose disease remained stable or progressed) had higher pre-treatment levels of miR-200c and miR-200b, and lower pre-treatment levels of miR-146a, compared to responders. High vs. low pretreatment levels of 9 miRNAs, including miR-200b, miR-200a, miR-429 and miR-200c, and pre/post treatment changes in 5 other miRNAs were associated with poor overall survival. ROC analyses showed that pre-treatment levels of miR-200b predicted patient death within 12 months with an AUC of 0.72 [185]. A follow-up study on plasma from additional docetaxel-treated CRPC patients did not confirm differences in pre-treatment levels of miR-200b, miR-200c or miR-146a between responders and non-responders [186]. The investigators also narrowed down the list of prognostic miRNAs to miR-132, miR-200a, miR-200b, miR-200c, miR-375 and miR-429 (i.e., high pretreatment levels associated with poor OS) [186].

A longitudinal study that compared plasma levels of miR-141 with PSA, CTC and LDH in 21 PC patients, most of whom were clinically progressing, found that patients with progressive disease showed an average increase in these four biomarkers, and patients with nonprogressive disease showed the opposite trend [187]. In a very recent study, high baseline plasma levels of miR-141 and miR-375-3p were shown to predict time to progression of metastatic CRPC patients treated with docetaxel (*N* = 40) or the androgen antagonist abiraterone (*N* = 44) [188]. High baseline levels of miR-141 and miR-221-3p were significantly associated with a shorter OS in the abiraterone-treated cohort, while high levels of miR-141 and miR-375-3p were significantly associated with shorter OS in the docetaxel-treated cohort; all these differences were highly statistically significant [188]. It is interesting to note that the levels of miR-141 decreased after one cycle of therapy with docetaxel and rose at the time of radiological progression of metastatic CRPC patients [188].

A recent study of a small cohort of patients with metastatic PC was focused on identifying plasma miRNAs and other parameters associated with development of resistance to the androgen antagonists abiraterone and enzalutamide [189]. Univariate analysis identified an association between plasma levels of miR-141 (high), miR-21-5p (low) and miR-223-3p (low), time to development of castration resistance (tCRPC) and low blood hemoglobin (Hb) levels with poor PFS and OS. miR-141 and miR-223-3p dropped out upon multivariate analysis, which indicated miR-21, Hb and tCRPC as independent factors predicting OS, and miR-21 and tCRPC predicting PFS [189].

The studies described above point toward circulating miR-200, in particular miR-141, as a marker of disease progression, but the utility of measuring miR-200 for diagnosis of early-stage PC is less clear. In their study of patients with localized PC, advanced PC and BPH, Zhang et al. did not detect a difference in serum miR-141 levels in patients with localized PC compared to BPH controls [183]. An analysis of a 4-miRNA panel in plasma from 59 PC patients, very few of whom had lymph node lesions or metastases, compared to 11 healthy controls and 16 BPH patients reported decreased levels of miR-141 and miR-375 in the PC group [190]. However, differences in miR-141 levels were not significant, and ROC curve analysis identified miR-375 as the best miRNA for discriminating between patients and controls [190]. Our laboratory compared plasma levels of a panel of 12 miRNAs, including miR-141, in 36 patients with localized PC and 31 urology patients whose prostate biopsy results confirmed the absence of PC, but did not identify differences in miR-141 among the 2 groups [191]. Two other studies of sera from PC patients of various stages and grades and healthy controls or BPH controls identified miR-141 [192] or miR-141 and miR-200b [193] among miRNAs that were upregulated in the PC patients, but did not perform comparisons between early-stage or low-grade disease subgroups and controls.

Quantification of a 9-miRNA panel that included miR-141 and miR-200c in whole plasma and plasma-derived extracellular vesicles from 50 PC patients with low or high Gleason scores and 22 patients with BPH identified miR-21-5p, miR-375 and miR-200c as having potential diagnostic significance; interestingly, miR-21-5p and miR-200c yielded stronger statistical values when analyzed in extracellular vesicles compared to whole plasma, and miR-375 performed best when analyzed in plasma [194].

A variety of DNAs, RNAs, proteins and small molecules have been proposed as urine markers for prostate cancer (reviewed in [195]). The abundance of prostate-derived molecules in the urine can be increased by massaging the organ during digital rectal examination (DRE). Different types of urine samples—unfractionated urine, cell-free urine supernatants, exosomes and sediment after centrifugation—have been studied.

Quantification of miR-141, miR-21-5p and miR-205-5p in unfractionated urine samples from patients with PC, bladder cancer or BPH and healthy controls indicated increased levels of all 3 miRNAs in both categories of cancer patients; it is noteworthy that miR-141 yielded the best ROC curve AUC for distinguishing PC patients from subjects with BPH, which is beyond the capabilities of PSA measurements [196]. A study that combined several patient cohorts analyzed 45 miRNAs in cell-free urine samples from patients with localized PC or BPH, and identified miR-141, miR-200a, miR-200b and miR-200c among upregulated miRNAs in the PC patients. However, further analyses excluded these miRNAs from the final 3-miRNA ratio model proposed by the investigators [197].

An analysis of miRNAs contained in the sediment fraction of centrifuged urine samples obtained after DRE that quantified 12 miRNAs, including miR-200c, did not identify differences in miR-200c levels in PC patients compared to tumor-free patients [198]. Quantification of a panel of miRNAs contained in urine EVs, clarified urine and plasma from patients with PC or BPH healthy controls identified urine EV as the most informative source of miRNAs to distinguish PC from the two control groups [199]. Calculations of miRNA ratios from the EV data revealed six miRNA pairs, including miR-200b, miR-30e, that identified the PC patients with high accuracy [199]. In a study aimed at identifying urine exosome-associated mRNAs and miRNAs for PC diagnosis, Davey et al. [200] used an affinity-capture method to isolate urine EVs from 56 subjects, 28 of whom were diagnosed with prostate cancer, and 28 with benign conditions that warranted prostate examination. The analysis yielded 4 miRNAs, including miR-141, with elevated levels in the PC group; however, miR-141 was not included in a panel of 2 miRNAs (i.e., miR-375 and miR-574) and 4 mRNAs proposed for PC diagnosis [200].

The role for urine miR-200 as prognostic markers of PC remains to be established. In a study aimed at identifying noninvasive markers to monitor low-risk PC patients, Zhao et al. analyzed a panel of methylated DNA markers and miRNAs, including miR-141, in DRE-urine samples from 103 patients with localized, Gleason Score 1 tumors. Results did not identify an association between miR-141 levels and disease progression [201].

### 4.5. Ovarian Cancer

Ovarian Cancer (OC) ranks #8 in terms of incidence and cancer mortality in females worldwide [106]. The majority of OC cases are derived from the epithelium, and are categorized into five histological subtypes termed high-grade serous, low-grade serous, endometrioid, clear-cell and mucinous epithelial OC, with the high-grade serous subtype representing about 75% of cases. At early stages, OC can be asymptomatic or paucisymptomatic, and is frequently misdiagnosed as irritable bowel disease. The lack of specific symptoms often delays diagnosis to advanced stage metastatic leading to the poor prognosis of this malignancy. The standard of care for patients with advanced OC is primary debulking surgery and systemic chemotherapy with carboplatin and paclitaxel, with the absence of residual tumor being the main factor predicting survival (reviewed in [202,203]).

Carbohydrate antigen-125 (CA-125) has been the biomarker of choice for the management of OC patients for many years. However, CA-125 has low sensitivity for diagnosis of early stages of the disease and it is affected by several physiological conditions, such as menstruation and pregnancy [204]. Searching for novel biomarkers for early diagnosis of OC, several studies evaluated the levels of miR-200 family members through liquid biopsy. Taylor and Gercel-Taylor were the first to assess the expression levels of several miRNAs in EpCAM^+^ exosomes isolated from women with benign ovarian disease or OC. The authors observed that the levels of miR-200a, miR-200b, miR-200c and miR-141 in tumor biopsies strongly correlated with the levels measured in exosomes [205]. Subsequent studies reported a significant increase in miR-200a, miR-200b and miR-200c in the serum of EOC patients compared to matched controls [206,207,208,209,210]. Plasma levels of all miR-200 family members were increased in OC patients compared to both healthy women and patients with non-malignant ovarian masses [211]. While the expression of miR-200a significantly correlated with histological subtype (mucinous and serous) and only marginally with stage and metastasis, high expression of miR-200c strongly correlated with stage III/IV disease and the presence of metastasis and lymph node invasion [207]. Kan and colleagues found that the combination of miR-200b and miR-200c was the best model to discriminate women with EOC from healthy women, with an AUC of 0.784 [206]. Similarly, Yokoi and colleagues [212] identified miR-200a among 8 serum miRNAs whose levels, combined with CA-125, distinguished OC patients from healthy controls with an AUC of 0.994, a sensitivity of 0.984 and a specificity of 0.956. Moreover, the authors found that 7 serum miRNAs, comprising miR-200a, discriminated early-stage OC from benign ovarian tumors with an AUC of 0.902, a sensitivity of 0.861 and a specificity of 0.833 [212]. As described for other epithelial malignancies, the increased circulating levels of miR-200 in EOC patients is in apparent contrast with their downregulation in EOC-infiltrated abdominal lymph nodes [213]. A possible explanation for these findings is that the expression of miR-200 might be downregulated upon EMT and invasion in the ascitic fluid, while the subsequent reorganization into metastatic colonies and reactivation of an epithelial differentiation program (MET) might be associated with increased expression of these miRNAs and possibly their release into the bloodstream of patients with metastatic OC. Analyses of ascitic fluid and peritoneal lavages from a small cohort of OC patients identified elevated expression of all of miR-200 in the patients compared to plasma samples from healthy post-menopausal women, and also indicated an association between high ascites-derived miR-200b levels and reduced OS [214]. By testing serum samples, Gao and Wu found that miR-200c and miR-141 can discriminate OC patients from controls. Interestingly, they also found that the expression of miR-200c decreases from stage I to stage III-IV, while miR-141 showed the opposite trend [215]. Patients with low expression of serum miR-200c and high expression of serum miR-141 had a significantly shorter overall survival [215]. Meng and colleagues observed that high levels of miR-429 in serum from 180 EOC patients positively correlated with CA-125 levels, advanced stage and shorter overall survival [216].

An assessment of miR-21, miR-34a, miR-200b and miR-205 along with CA-125 in plasma samples from 51 OC patients and 50 controls (healthy or with benign pelvic lesions) revealed miR-200b as the only miRNA with significantly elevated levels in the patients, albeit with broad variability and lack of correlation with CA-125 levels [217]. Analyses of plasma samples obtained from 33 patients after treatment (chemotherapy only or surgical debulking followed by adjuvant chemotherapy with or without neoadjuvant chemotherapy) indicated a borderline association between a post-treatment increase in miR-200b and shortened PFS; initial CA-125 values and their alterations post-treatment did not associate with PFS [217].

A comparison of 163 EOC patients with healthy women revealed higher exosomal miR-200a, miR-200b and miR-200c levels in the cancer patients, which correlated (except for miR-200a) with advanced-stage lymph node-positive disease and shorter overall survival [218]. Another study confirmed increased levels of exosomal miR-200b in OC patients compared to healthy controls, and linked high exosomal miR-200b to increased CA-125 levels and poor overall survival [219]. Exosomal miR-200 present in the serum of OC patients could derive from epithelial cancer cells undergoing MET as they establish novel metastatic sites. It is also possible that release of exosomes containing miR-200 by ascitic OC cells might decrease their intracellular levels, resulting in increased invasive potential.

A recent study employed plasma and serum samples from 118 EOC patients and 96 healthy controls at institutions in the U.S. and Hong Kong-China [220]. Analysis of all 5 miR-200 family members in serum/plasma samples revealed increased levels of miR-200a, miR-200b and miR-200c in the OC patients compared to controls. Separate analyses of data from the two cohorts revealed interesting differences possibly reflecting their distinct ethnicities: for example, while samples from U.S. EOC patients showed increased levels of miR-200a, miR-200b, miR-200c and miR-141, those from the Hong Kong-China cohort showed increased levels of miR-200b and miR-429, and decreased levels of miR-141. The investigators also found subtype-related differences in miR-200 levels, with serous and mucinous subtypes showing increased levels of miR-200b and miR-200c, and clear-cell and endometrioid subtypes showing increased miR-429 levels (based on the Hong Kong-China cohort, which had the highest number of different subtypes). Neural network modeling identified miR-200a,200b,429,141 as the best model to discriminate cancer patients and controls in the U.S. cohort, with an AUC of 0.904, while miR-200b,200c,429,141 was the best model to distinguish cancer patients and controls in the Hong Kong-China cohort, with an AUC of 0.901 [220].

Some OC patients respond to treatment with the anti-VEGFA monoclonal antibody bevacizumab alone or together with standard chemotherapy. A search for plasma miRNAs associated with response to bevacizumab plus standard chemotherapy (carboplatin and paclitaxel) compared to standard chemotherapy alone in 116 patients from clinical study ICON7 [221] indicated that low plasma levels of miR-200c may predict response to the drug combination [222]. Analyses of data from all 116 patients without grouping according to treatment regimen showed that high plasma levels of miR-141 and miR-200b were each associated with shorter PFS [222].

### 4.6. Bladder Cancer

Bladder Cancer (BCa) is the tenth most frequently diagnosed cancer worldwide, and is about four times more common in males than in females [106]. BCa most frequently originates from the urothelium, the epithelial layer that covers the inner surface of the bladder. Tumors that invade the detrusor muscle (muscle-invasive bladder cancer, MIBC) are more likely to metastasize to lymph nodes or other organs. Approximately 75% of newly diagnosed patients have non-muscle-invasive bladder cancer (NMIBC) and 25% have MIBC or metastatic disease [223,224,225]. The majority of patients with NMIBC can be successfully treated with transurethral surgery followed by intravesical chemotherapy or immunotherapy, but are nevertheless at risk of disease recurrence and must be monitored by periodic cystoscopy and urine cytology. MIBC is treated by radical cystectomy and chemotherapy, but prognosis is poor, with a 5-year survival rate of less than 5% in patients with metastatic disease. While patients with advanced disease frequently present with hematuria, this sign is often not evident in early-stage BCa, leading to a delay in diagnosis.

The need for long-term monitoring place a heavy burden on BCa patients and health care systems, and many studies have been aimed at identifying liquid biopsy biomarkers to facilitate BCa diagnosis and follow-up (reviewed in [226,227]).

A small number of studies employed serum or plasma as sources of biomarker miRNAs for BCa diagnosis/prognosis. Microarray analysis to detect miRNAs in cell-free plasma samples from 20 BCa patients and 18 controls (not well balanced in terms of gender or age) identified 79 differentially expressed miRNAs, including miR-200b, which was more abundant in the patients’ plasma, but this study did not include a validation step [228]. A two-step investigation of serum samples measured a panel of miRNAs that included miR-141 and miR-200b and detected increased levels of miR-141 and miR-639 in the small discovery cohort, but these differences were not confirmed in the validation cohort [229]. Another two-step study that employed RNA sequencing in the discovery step did not identify miR-200 members among differentially expressed miRNAs [230].

Given the direct and prolonged contact of BCa cells with urine, this biofluid would seem to be the ideal source of biomarkers. However, as described below, a urine miRNA signature for BCa has not yet emerged. Several factors peculiar to urine samples may contribute to this lack of consensus, including differences in urine concentration, presence/extent of hematuria and differences in sample type (unfractionated urine, post-centrifugation sediment or supernatants or extracellular vesicles).

The first search for miRNAs in urine of BCa patients proposed a ratio of miR-126/miR-152 as a disease marker and did not report data for miR-200 [231]. Yun et al. [232] carried out a series of trial assays that confirmed the high stability of miRNAs in cell-free urine samples and then analyzed levels of miR-145 and miR-200a in cell-free urine from a cohort of 207 patients with primary BCa and 144 healthy controls. They observed decreased abundance of both miRNAs in patients with NMIBC and MIBC compared to controls (*p* < 0.001). ROC curve analyses identified miR-145 as a potential diagnostic marker for NMIBC and MIBC, with AUCs of 0.729 and 0.790, respectively. Multivariate analysis identified miR-200a as a predictor of NMIBC recurrence, and a Kaplan–Meier analysis showed that NMIBC patients with lower miR-200a levels were at greater risk of disease recurrence [232].

An analysis of urine sediment from 51 BCa patients and 24 controls identified reduced levels of miR-200a, miR-200b, miR-200c, miR-141, miR-429, miR-192 and miR-155 in the patients’ samples compared to the controls, independent of tumor stage or grade [233]. However, in contrast to observations made by Yun et al., analysis of the cell-free urine fraction did not reveal differences in levels of miR-200a and other miR-200 family members among patients and controls. The authors pointed out an inverse correlation between the downregulation of the miR-200 family and the expression of EMT markers (e.g., ZEB1 mRNA) in the urine sediment [233]. Comparison of miRNA levels in 9 patients before and after tumor resection revealed a post-surgery increase in all of the miR-200 members in both the urine sediment and supernatant fractions [233]. In additional analyses of urine sediment, miR-200 members either were not reported as differentially expressed in BCa samples [234,235] or more abundant (miR-200a, 200b, 200c and miR-141) in BCa but they did not pass a univariate test for association with BCa [236].

In a more recent study carried out on unfractionated urine samples from 66 male BCa patients and 53 age and sex-matched controls, NeKoohesh et al. [237] detected a decrease in miR-141 levels in the patient group, which however was not statistically significant; miR-141 was not linked to clinicopathogical parameters such as stage, grade or recurrence, although increased levels were measured in patients with opium addiction, which represented a possible risk factor in the cohort [237].

Sapre et al. [238] performed a two-step discovery-validation study on unfractionated urine from a total of 131 subjects comprising BCa patients with active disease (initial diagnosis or recurrence), patients with a history of BCa but no recurrence, and healthy controls. Starting out with a panel of 12 candidate miRNAs, the study yielded a 6-miRNA panel composed of miR-200c, miR-16, miR-205, miR-21, miR-221 and miR-34a (all more abundant in the active disease group) that predicted patients with active disease with an AUC of 0.85 in the discovery step and 0.74 in the validation step [238].

A study by Du et al. [239] that started out with RNA sequencing of urine supernatants from 6 BCa patients and 6 controls followed by several validation steps on large numbers of samples identified a 7-miRNA panel that included miR-200a (together with miR-7-5p, miR-22-3p, miR-29a-3p, miR-126-5p, miR-375, and miR-423-5p) that distinguished between BCa patients and controls with an AUC of 0.916 in the validation set [239]. miR-200a and miR-423-5p were down-regulated in BCa (although the ROC analysis for miR-200a yielded a weak AUC value of 0.692). Kaplan–Meier analysis showed that low urine miR-200a levels and high urine miR-22-3p values predicted a short relapse-free survival (RFS) interval in NMIBC patients, and multivariate Cox regression analysis revealed that the two miRNAs were independently associated with RFS in NMIBC. None of the analyzed miRNAs predicted disease recurrence in MIBC patients [239]. Another study of urine supernatant that started out with an RNA-sequencing step identified 3 potential signatures for distinguishing MIBC, NMIBC Grades 1 + 2, and NMIBC Grade 3 cases from controls and reported upregulation of miR-200c and 7 other miRNAs in NMIBC Grade 3 patients [240].

In a recent study aimed at identifying urine miRNAs of prognostic value for BCa, we measured levels of a panel of miRNAs in cell-free urine from a prospective cohort of 32 high-risk patients, 31 low-risk patients and 37 healthy controls. A series of trial assays led to selection of candidate miRNAs that were not affected by urine specific gravity and hemolysis. Although none of the 14 tested miRNAs differed between low-risk patients and controls, 6 miRNAs—miR-21, miR-34a, miR-193a, miR-141, miR-200a and miR-200c—were identified as upregulated in high-risk patients compared to low-risk patients using stringent selection criteria (fold change > 2.5, Bonferroni-adjusted *p* values < 0.005). Interestingly, the levels of these miRNAs in plasma samples from the study subjects were very low and did not differ among the three groups. Further analyses led to construction of a two-step decision tree using urine miR-34a, miR-193a and miR-200a, with normalization against miR-125b, that accurately classified high-risk and low-risk patients. Cox proportional hazards regression analysis identified miR-200c among 6 miRNAs whose elevated levels were associated with shorter event-free survival [241].

## 5. Conclusions and Perspectives

Mechanistic studies of miR-200-regulated networks have revealed the role of these miRNAs as central hubs of two-node double negative feedback loops (DNFL) in which transcription factors (TF) suppress expression of miR-200 which, in turn, negatively regulate the TF (Figure 2). Interestingly, DNFL generate bi-stable switches that confer robustness and plasticity to critical regulatory circuits which, in response to variation of biological parameters beyond a threshold value, abruptly switch between on- and off-states [242]. Collected evidence converges over a central position of the miR-200 family in at least 4 DNFL (Figure 2) controlling the EMT-MET switch (ZEB/SNAIL), as well as two pervasive shapers of the cell’s transcriptomic landscape (Myc and SIRT1), which play a key role in the progression towards metastatic dissemination. As metastatic tumors remain essentially incurable to date, future studies should aim at an integrated understanding of these regulatory networks, with the final goal of developing novel therapeutic tools that target the invasive growth program through the rewiring of these central hubs directing EMT/MET and cancer stem cell plasticity.

Although some studies indicate a possible role of high levels of miR-200 as circulating biomarkers of cancer progression (see Table 1), this is in apparent contrast with the well-established role of these miRNAs as EMT-inhibitors and some discrepancy in the outcome of different studies still persists. As described in this review, EMT is a highly plastic process in which cells from the invasive front of primary tumors acquire mesenchymal features needed for metastatic invasion, while the metastatic colonies of seeded tumor cells need to switch from EMT to MET program. Thus, the differences in the expression of the miR-200 family members may be highly dependent on the area of the tumor examined (invasive front vs. core of primary tumors or metastatic colonies). These considerations suggest that determination of miR-200 expression in tumor biopsies may provide information that is of excessive complexity for diagnostic/prognostic purposes. Some inconsistencies among studies of circulating miR-200 are likely to result from the lack of a common standard procedure for the pre-analytical and analytical steps employed in these analyses. This problem is particularly important when analyzing urine samples, which may vary widely in terms of concentration, presence of RNases, contamination with cellular sediments and presence of hematuria.

Quantification of miRNAs in ml size samples of biofluids with large total body volumes such as blood and urine depend on the sensitivity of the detection technique and the possibility to concentrate the sample while maintaining the components of interest. Leung et al. [243] recently described a sensitive method to detect miR-141 in urine that combines tagged probes that generate an electrochemical signal in the presence of the target miRNA. This dual-probe technique and further innovations in sample processing and miRNA detection will strengthen the prospect of translating experimental observations to the application of miR-200 family members as noninvasive biomarkers to the clinic.

## Figures and Tables

**Figure 1 cancers-13-05874-f001:**
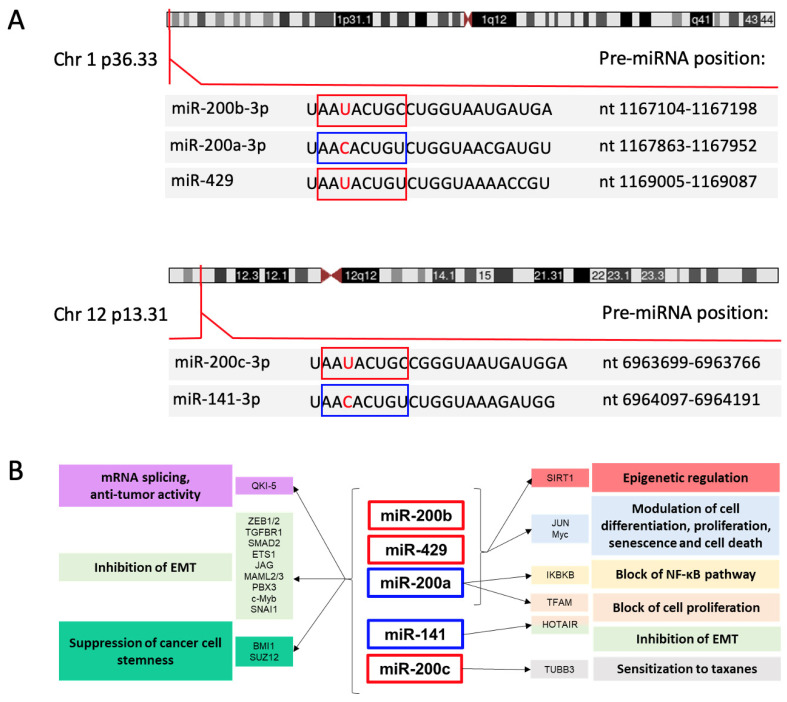
Genetic organization of the miR-200 family in humans. (**A**) Indicated are the chromosome loci encoding the two clusters of the miR-200 family, the positions of the pre-miRNA hairpins, and the sequences of the mature miRNAs with seed sequences highlighted in boxes. The seed sequences differ by one nucleotide (indicated in red). As described in the text, there is a strong bias for production of the -3p products from the pre-miRNAs. Information is from the Sanger miRbase (https://www.mirbase.org, accessed on 28 July 2021) [12] and https://genome.ucs.edu/index.htlm (accessed on 11 October 2021) University of California Santa Cruz Genome browser. (**B**) Schematic representation of the main targets and pathways controlled by the miR-200 family members grouped according to their genomic clusters. Red and blue boxes indicate the miRNAs with the two different seed sequences. A selection of target genes and corresponding biological effects are indicated using matching colors.

**Figure 2 cancers-13-05874-f002:**
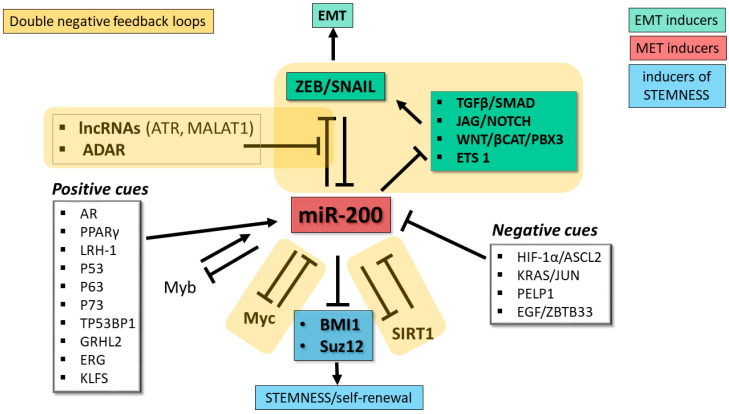
Regulatory networks involving miR-200 family. miR-200 family members play a key role as repressors of factors driving EMT (green boxes) and stemness (blue box), two key hallmarks of tumor progression and invasive growth. The miR-200 family lies at the core of multiple double negative feedback loops (DNFL, indicated by yellow shaded areas) controlling EMT plasticity: one direct with the ZEB and SNAIL transcription factors, the other indirect through signaling pathways that control ZEB and SNAIL. Additional miR-200-based DNFL control Myc and SIRT1, two pervasive regulators of the cell’s transcriptional landscape. White boxes indicate major positive and negative cues controlling miR-200 expression. EMT, epithelial-mesenchymal transition. MET, mesenchymal-epithelial transition.

**Table 1 cancers-13-05874-t001:** The clinical value of circulating miR-200 family members in breast, lung, colorectal, ovarian, prostate and bladder cancers. For studies that included discovery and validation steps, only the validation data are reported.

Tumor Type	Patients/Controls (Numbers)	Sample Type	Results	Proposed Application	Reference
Breast Cancer (BC)	BC (50) Healthy controls (50) Benign breast disease (20) Ovarian cancer (20)	Serum	↑ miR-200b, miR-200c and miR-429 in BC ↓ miR-200a in BC	Diagnosis	[124]
	BC patients (57; stage I-IV) Age-matched controls (20)	Plasma	↑ miR-141 and miR-200c in BC	Diagnosis	[125]
	Metastatic BC (193) Healthy controls (76)	CTC from plasma	↑ miR-200a, miR-200b and miR-200c in metastatic BC	Prognosis	[128]
	Metastatic BC (569)	Plasma	↑ miR-200 family in metastatic BC	Prognosis	[132]
	Invasive breast ductal carcinoma (78)	Plasma	↑miR-200b in node-positive patients and in patients with distant metastases	Prognosis	[111]
	BC (105)	Serum	↑ miR-141 in patients with brain metastasis	Prognosis	[130]
	Localized BC (50) Metastatic BC (25) Healthy controls (50)	Plasma	↑ miR-141 in patients with metastatic BC	Prognosis	[131]
	Early BC (133)	Plasma	↑ miR-200c in early-relapse patients	Prognosis	[133]
	Early BC (133) Metastatic BC (110)	Plasma	↑ miR-200b and miR-200c in metastatic BC	Prognosis	[134]
	BC (96) Healthy controls (14)	Serum	↑ miR-141 in BC	Prognosis	[135]
Non-small cell lung cancer (NSCLC)	NSCLC (100) Healthy controls (58)	Serum	↑ miR-200b in NSCLC	Diagnosis	[145]
	NSCLC (64) Healthy controls (58)	Sputum	↑ miR-200b in NSCLC	Diagnosis	[146]
	NSCLC (LUAD, SCC, 72) Healthy controls (50)	Plasma	↑ miR-141 in NSCLC	Diagnosis	[147]
	Longitudinal assessment of immunotherapy treated LUAD, SCC: responders (17); non-responders (17)	Serum	↑ miR-200 in responders	Predictive	[151]
	LUAD (50) Lung granulomas (30) Healthy controls (25)	Plasma-derived exosomes	↑ miR-200b in NSCLC	Diagnosis	[6]
	LUAD (18) Benign lung disease (18)	Pleural effusion-derived exosomes	↑ miR-141, miR-200b and miR-200c in NSCLC	Diagnosis	[148]
Colon-rectal cancer (CRC)	CRC (78) Healthy controls (86) CRC pre- and post-surgery (21)	Plasma	↑ miR-200c in CRC ↓ miR-200c after surgery	Diagnosis	[160]
	CRC (74 stage II-IV) Healthy controls (32)	Plasma	↓ miR-200c in CRC	Diagnosis	[161]
	CRC (156 Stage I-II III IV) Matched controls (156)	Plasma	↑ miR-141 in stage IV CRC	Prognosis	[162]
	CRC (187 Stage I-II III IV) Healthy controls (47)	Plasma	↑ miR-141 and miR-200b in stage IV CRC	Prognosis	[163]
	CRC (30) Colonic polyps (30) Healthy controls (30)	Serum	↑ miR-141 in CRC	Diagnosis	[164]
	CRC (182 Stage I-II III IV) Healthy controls (24)	Serum	↑ miR-200c in stage IV CRC	Prognosis	[167]
	Non-metastatic CRC (309) Metastatic CRC (234)	Plasma	↑ miR-141, miR-200a and miR-200b in metastatic CRC	Prognosis	[168]
	CRC without liver metastases (54)CRC with liver metastases (54)	Serum	↑ miR-141, miR-200a and miR-200b in metastatic CRC	Prognosis	[169]
	Resected CRC (50)	Plasma and exosomes from mesenteric vein (MV) and peripheral vein (PV)	↑ miR-200 family MV vs. PV	Prognosis	[172]
Prostate Cancer (PC)	Metastatic PC (25) Healthy controls (25)	Serum	↑ miR-141 in metastatic PC	Diagnosis	[102]
	Localized PC (14) Bone-metastatic PC (7) High-risk and intermediate-risk PC (71)	Serum	↑ miR-141, miR-200b and miR-200c in metastatic PC ↑ miR-141 in high-risk PC	Prognosis	[177]
	Localized low-risk PC (28) Localized high-risk PC (30) Metastatic CRPC (26)	Serum	↑ miR-141 in metastatic PC vs. localized low-risk PC	Prognosis	[178]
	Localized/locally advanced PC (26) Metastatic PC (25) Healthy controls (20)	Plasma	↑ miR-141 in metastatic PC vs. localized/locally advanced PC	Prognosis	[179]
	Metastatic CRPC (21) Healthy controls (20)	Serum	↑ miR-141 in metastatic CRPC	Diagnosis	[180]
	Localized PC (25) Metastatic CRPC (25)	Plasma	↑ miR-141	Diagnosis/Prognosis	[181]
	Localized PC (55) Metastatic PC (16) Recurrent metastatic PC (47) Non-recurrent PC (72)	Plasma vesicles Serum vesicles	↑ miR-141 and miR-200b in metastatic CRPC ↑ miR-141 in metastatic CRPC	Prognosis	[182]
	Localized PC (20) Metastatic PC (30) BPH (6)	Serum	↑ miR-141 in metastatic PC vs. localized PC or BPH	Diagnosis	[183]
	PC (102) Healthy controls (50)	Plasma	↑ miR-200b, miR-200c in PC	Diagnosis/Prognosis	[184]
	CRPC in docetaxel treatment (97) (Phase I)	Plasma/serum	↑ miR-200c and miR-200b associated with poor response to therapy ↑ miR-200a, miR-200b, miR-200c and miR-429 associated with poor OS	Prediction of docetaxel chemotherapy outcome	[185]
	CRPC in docetaxel treatment (89) (Phase 2)	Plasma	↑ miR-200a, miR-200b, miR-200c and miR-429 associated with poor OS	Prognosis	[186]
	Metastatic PC (21) clinically progressing and non-progressing	Plasma	↑ miR-141 in clinically progressing PC	Prognosis	[187]
	Metastatic CRPC treated with docetaxel (40) or abiraterone (44)	Plasma	↑ miR-141 in progression and associated with poor OS regardless of therapy	Prognosis	[188]
	PC (72) Healthy controls (34)	Serum	↑ miR-141 in PC	Diagnosis	[192]
	PC (31) BPH (31)	Serum	↑ miR-141 and miR-200b in PC	Diagnosis	[193]
	PC (50) BPH (22)	Plasma vesicles	↑ miR-200c in PC	Diagnosis	[194]
	PC (23) BPH (22) Healthy controls (20)	Urine	↑ miR-141 in PC	Diagnosis	[196]
	PC (758) BPH (289)	Urine	↑ miR-200a, miR-200b, miR-200c and miR-141 in PC	Diagnosis	[197]
	PC (10) BPH 8) Healthy controls (11)	Urine vesicles	↑ miR-200b/miR-30e in PC	Diagnosis	[199]
	PC (28) Benign controls (28)	Urine exosomes	↑ miR-141 in PC	Diagnosis	[200]
Ovarian Cancer (OC)	OC (50) Benign ovarian adenoma (10) Healthy controls (10)	Serum	↑ miR-141, miR-200a, miR-200b and miR-200c in OC	Diagnosis	# [205]
	OC (28) Healthy controls (28)	Serum	↑ miR-200a, miR-200b and miR-200c in OC	Diagnosis	[206]
	OC (70) Healthy controls (70)	Serum	↑ miR-200a, miR-200b and miR-200c in OC	Diagnosis/prognosis	[207]
	OC (8) Benign cystadenoma (5)	Blood	↑ miR-200a, miR-200b and miR-200c in OC	Diagnosis/prognosis	[208]
	OC (95) Benign pelvic mass (95)	Plasma	↑ miR-200c in OC	Diagnosis	[209]
	OC (9) Benign ovarian tumor (7)	Plasma	↑ miR-200a, miR-200b and miR-200c in OC	Diagnosis	[210]
	OC (28) Benign ovarian mass (12) Healthy controls (60)	Plasma	↑ miR-141, miR-200a, miR-200b, miR-200c and miR-429 in OC	Diagnosis	[211]
	OC (185) Benign ovarian tumor (43) Healthy controls (63)	Serum	↑ miR-200a in OC	Diagnosis	[212]
	OC (26) Healthy controls (34)	Ascites and peritoneal lavages (patients) Plasma (controls) (healthy women)	↑ miR-141, miR-200a, miR-200b, miR-200c and miR-429 in OC	Diagnosis Prognosis (only miR-200b)	[214]
	OC (93) Healthy controls (50)	Serum	↑ miR-141 and miR-200c in OC (not in borderline OC) ↓ miR-141 and ↑ miR-200 (metastatic vs. non-metastatic)	Diagnosis/prognosis	[215]
	OC (180) Healthy controls (66)	Serum	↑ miR-429 in OC	Diagnosis/prognosis	[216]
	OC (51) Benign pelvic tumor (25) Healthy controls (25)	Plasma	↑ miR-200b in OC	Diagnosis/prognosis	[217]
	OC (163) Benign ovarian tumor (20)	Serum	↑ miR-200a, miR-200b and miR-200c in OC	Diagnosis Prognosis (only miR-200b and miR-200c)	[218]
	OC (106) Benign cystadenoma (8) Healthy controls (29)	Plasma	↑ miR-200b in OC	Prognosis	[219]
	OC (118) Healthy controls (112)	Blood/plasma/serum	↑ miR-141, miR-200a, miR-200b and miR-200c in OC (U.S. cohort) ↓ miR-141 ↑ miR-200b and miR-429 In OC (Hong Kong-China cohort)	Diagnosis	[220]
	OC (116)	Plasma	miR-141 high vs. low (worse OS) miR-200b high vs. low (worse OS) miR-200c high vs. low (worse response to bevacizumab)	Prognosis Prediction of response to treatment	[222]
Bladder Cancer (BCa)	BCa (20 NIMBC + MIBC) Healthy controls (18)	Plasma	↑ miR-200b in MIBC	Diagnosis	[228]
	BCa (207 NIMBC + MIBC) Healthy controls (144)	Urine supernatant	↓ miR-200a in NIMC and MIBC	Diagnosis	[232]
	BCa (51) Healthy controls (24)	Urine sediment/supernatant	↓ miR-200 family in BCa (sediment) ↑ miR-200 family in post-surgery BCa	Diagnosis	[233]
	BCa (50) Cancer-free patients (25)	Urine	↑ miR-200c in BCa	Diagnosis	[238]
	BCa (63) Healthy controls (63)	Urine supernatant	↓ miR-200a in BCa	Diagnosis	[239]
	BCa (46 MIBC, NMIBC G1 G2 G3) Healthy controls (14)	Urine	↑ miR-200a in NIMBC G3	Risk-Stratification	[240]
	BCa (63 high- and low-risk) Healthy controls (37)	Urine	↑ miR-200a and miR-200c in high-risk BCa	Risk-Stratification	[241]

The simbol ↑ indicates increased levels, the symbol ↓ indicates decreased levels. The symbol # indicates a single study that employed microarrays for miRNA detection; all other studies employed qRT-PCR. BC = breast cancer, CRC = colorectal cancer, NSCLC = non-small cell lung cancer, LUAD = lung adenocarcinoma, SCC = squamous cell carcinoma, OC = ovarian cancer, CRPC = castration-resistant prostate cancer; BPH = benign prostatic hyperplasia, NMIBC = non-muscle invasive bladder cancer, MIBC = invasive bladder cancer.

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
