# Peer review of "The miR-200 Family of microRNAs: Fine Tuners of Epithelial-Mesenchymal Transition and Circulating Cancer Biomarkers"

_cancers, 2021, doi:10.3390/cancers13235874_

Round 1
Reviewer 1 Report
This manuscript reviews miR-200 family functions in epithelial-to-mesenchymal transition (EMT) and the potential utility of circulating miR-200 family members as biomarkers. It nicely covers this field and is well written.
You need to change “miR-200” to “miR-200 family” where this can be applicable, including the titles of sections and Fig. 2 to be precise. For example, section “2. miR-200 and the epithelial-mesenchymal transition” includes descriptions of miR-200b/200a/429 and miR-141. Table 1 is missing. Fig.1 legend is in the main text. Fig.1 requires an elaborate depiction of the function of each miRNA separately with each miR name. The section “4. Circulating miR-200 as biomarkers” contains plenty of information and comprises the main part of this manuscript. It requires a figure and/or table for this section for easy understanding.Full names of PFS and OS should be indicated.
Author Response
We thank the Reviewer for the insightful comments. We extensively modified the manuscript, with revisions highlighted in yellow. We hope that the Reviewer will find it acceptable for publication.

Reviewer 2 Report
Authors comprehensively describe the literature related to the miR-200 family members. Two main sections can be distinguished, one explaining the importance of miR200 on their contribution to the EMT phenotype, drug resistance and stemness, and the other one describing their potential as circulating biomarkers for cancer diagnosis.
I would mention one main obstacle when reading the article: despite well written and of scientific relevance, too much descriptive work is shown in the text that difficults the reader to have a fluent understanding of the core of these both parts. I would recommend the authors generate two tables:
1- one for the miRNAs-target interaction, function, direction in regulating EMT phenotype, etc. described in section 2 - starting from line 100. (including the references in the table from which the discoveries were taken)
2- the other table describing miRNAs in section 4, maybe to summarize the miRNAs described for each cancer type and their main function, axis and body fluid where they were found (including the references in the table from which the discoveries were taken)
Minor changes:
Line 27- miRNAs mainly bind to 3'UTR as stated by the authors, but they can also bind to 5'UTR and coding sequence regions, which can potentially regulate protein levels. Authors should briefly comment on that.
Line 36- comment which are the biological implications of miRNA-5p forms for having lower percent identity.
Line 37- Not every audience is expert on the Riken Fantom 5 database, I would recommend authors briefly explain that.
Line 45- series of miRNAs are described as miR-200b-200c-492 and miR-200a-miR-141. If authors want to add or not "miR" in front of every single miRNA they should be consistent with it.
Line 50 to 56 - authors briefly mention distinct regulation of miRNA expression, and I would appreciate if they can comment more on the different mechanisms of regulating their expression they propose with more examples.
Line 65 through Line 71 - I would suggest the text comprised between these lines has to be the figure legend of figure 1.
Line 83 through line 85 - This is a very long sentence and with complicated reading as it is. One suggestion: The high plasticity of these changes enables cancer cells to switch between gene programs and EMT states, which can be driven by ligand-receptor interactions ...{ADD refs}
Line 86 - add references to the ligand-receptor interactions
Line 159- The authors state that reintroduction of miR200 members reversed the drug resistance associated with EMT, but need to prove that with experimental data references, or reduce the strength of the sentence per se.
Line 162- This is just a suggestion but this last paragraph before moving to section 3, could be relocated to the beginning of section 4, pinpointing that not only the secretion of miRNAs can be potential biomarkers but also highlighting that secreted miRNAs could function as regulatory molecules to distant cells/tissues, just like the example authors mention in lines 162-169
Line 170- in section 3, authors should have a introductory paragraph explaining the importance of describing the knowledge about the upstream pathways that regulate the expression of miR-200 members, at the biological level and also why in several types of diseases, it is known that alterations in other pathways lead to miR200 family deregulation.
Is there a commonality between all the factors that regulate miR200 expression? ZEB1, 2, NANOG, HIF1A, ASCL2, DNMT1, MYC, DNMT3A, FERMT2, KDM5B, PELP1, ZBTB33, EGF, KRAS, JUN, SP1, ERG, TMPRSS2, KLF5, c-Myb, p63, p73, TP53BP1, PPARa, LRH-1, SHP, and lncRNAs, DNA metilation and RNA deamination, among others, are a number of factors that the authors suggest as important regulators of miR200 family members expression. Despite depicted in Figure 2, it would be good if authors could do a small pathway analyses with all Known upstream regulatory elements of miR-200 to identify biological processes responsible of miR200 alterations. That would give a more comprehensive overview of which molecular processes or disease states orchestrate their expression.
Line 311 through line (mid)-320- Authors should consider this part of the paragraph as an introductory paragraph for section 4.
Line 320-line 324 - it is IMPERATIVE that authors revise the whole body of the text since from line 320 to line 324 there is a piece of text that was copied from other manuscript or template. It explains part of the manuscript guidelines for authors, to my understanding. This text should be removed promptly.
Lines 325 to 334 - authors should consider this part of the text as the legend for Figure 2.
Line 335 - as mentioned before, authors could reduce a bit the extension of this whole section including a table that depicts the main miRNAs for each tumor type, biological function, direction of expression during disease, targets (if known), phenotype/survival when up/down, body fluid(s) where were found, technique, number of samples, and reference(s).
Authors may consider to include these references in their manuscript:
- DOI: 10.1016/j.yexcr.2018.04.024
- DOI: 10.1016/j.tranon.2021.101228
- DOI: 10.1186/s12935-021-01784-4
- DOI: 10.3389/fonc.2020.526850
- DOI: 10.20517/cdr.2020.106
- DOI: 10.1016/j.tranon.2021.101228
- DOI: 10.1038/onc.2010.201
- https://ar.iiarjournals.org/content/anticanres/35/3/1401.full.pdf - paper that supports this family as an oncomiR.
- DOI: 10.1073/pnas.1212769110
- DOI: 10.1016/j.omtn.2019.06.015
Author Response

(The authors gave the same response as above.)

Round 2
Reviewer 2 Report
Authors have reviewed the text accurately and followed the suggestions of the reviewer. I still consider the hTFtarget analysis the authors did of great importance but I understand it might make the whole story more difficult to understand.
English was improved and several references were added, so now manuscript can be considered for publication.
Author Response
We thank the Reviewer for the kind comments. We have include a few minor changes to correct minor grammatical errors , added one missing reference and legend to Fig. 1B.
The revised manuscript is now uploaded.